# Earth Observation-Informed Risk Maps of the Lyme Disease Vector *Ixodes scapularis* in Central and Eastern Canada

**Serge Olivier Kotchi** [1,2]**, Catherine Bouchard** [1,2,*]**, Stéphanie Brazeau** [1,2] **and Nicholas H. Ogden** [1,2]

1    Public Health Risk Sciences Division, National Microbiology, Public Health Agency of Canada, Saint-Hyacinthe, QC J2S 2M1, Canada; serge-olivier.kotchi@canada.ca (S.O.K.); stephanie.brazeau@canada.ca (S.B.); nicholas.ogden@canada.ca (N.H.O.)
2    Groupe de Recherche en Épidémiologie des Zoonoses et Santé Publique (GREZOSP), Faculté de Médecine Vétérinaire, Université de Montréal, Saint-Hyacinthe, QC J2S 2M1, Canada
*    Correspondence: catherine.bouchard@canada.ca

**Abstract:** Climate change is facilitating the geographic range expansion of populations of the tick vector of Lyme disease *Ixodes scapularis* in Canada. Here, we characterize and map the spatio-temporal variability of environments suitable for *I. scapularis* using Earth observation (EO) data. A simple algorithm for *I. scapularis* occurrence (cumulative degree-days and forest: CSDF) was developed by combining cumulative annual surface degree-days above 0 °C and forest cover. To map the environmental risk of *I. scapularis* (risk of *I. scapularis*: RIS) in central and eastern Canada from 2000 to 2015, CSDF was adjusted using data from an *I. scapularis* population model. CSDF was validated using cumulative annual degree days >0 °C (CADD) from meteorological stations, and CSDF was strongly associated with CADD ($n = 52$, $R^2 > 0.86$, $p < 0.001$). Data on field surveillance for *I. scapularis* ticks (2008 to 2018) were used to validate the risk maps. The presence of *I. scapularis* ticks was significantly associated with CSDF, and at a limit of 2800, sensitivity approached 100%. RIS increased over the study period, with the highest values in 2012 and the lowest in 2000. The RIS was on average higher in Ontario and Quebec compared to other provinces, and it was higher in the southern parts of the provinces. The proportion of the populated areas with a positive RIS increased on average in central and eastern Canada from 2000 to 2015. Predicted *I. scapularis* occurrence identifies areas with a more probable risk of tick bites, Lyme disease, and other *I. scapularis*-borne diseases, which can help guide targeted surveillance, prevention, and control interventions.

**Keywords:** *Ixodes scapularis*; Lyme disease; risk maps; cumulative surface degree days; spatio-temporal variations; MODIS temperature and land cover products; earth observation; climate change

## 1. Introduction

Lyme disease, caused by bacteria of the *Borrelia burgdorferi* sensu lato complex, is the most common vector-borne disease in the temperate zone of the northern hemisphere. In eastern North America, the vector is the tick *Ixodes scapularis*, which is expanding its geographic range northward into and through southeastern and south-central Canada [1,2]. Lyme disease cases in Canada are increasing commensurate with northward expansion of the range of the tick into three main zones—southern Manitoba and adjacent northwestern Ontario, southern Ontario and Quebec, and the maritime provinces of New Brunswick and Nova Scotia [2–5]. There is compelling evidence that recent climate warming in southern Canada has facilitated range expansion of the tick [6]. Future projected climate change is expected to drive further range expansion of *I. scapularis* within Canada and thereby to be accompanied by an increase in the human population exposed to ticks and an increase in the number of human cases of Lyme disease [7,8]. In 2017, there were 2025 cases reported to the Public Health Agency of Canada (PHAC) compared to 144 cases in 2009 [5]. It is estimated that by 2020, 80% of the population of eastern Canada will live in areas with established tick populations [2]. If a person contracts Lyme disease, those with an early

localized infection may present with an *erythema migrans* rash sufficient for diagnosis and can be treated with a 2- to 3-week course of antibiotics. However, if left untreated, the infection can progress to disseminated illness with multiple cutaneous lesions and neurologic (radiculopathy, cranial neuropathy, meningitis), cardiac (heart block), and/or joint involvement.

Most people who contract Lyme disease do so from locations where vector populations are established, although a small number of cases may be acquired by adventitious ticks dispersed out of tick populations by migratory birds [9]. Therefore, identifying where and when environmental conditions are suitable for populations of the tick identifies the entomological risk from Lyme disease. Bouchard et al. (2015) reviewed Canadian studies that connect the spatio-temporal pattern of Lyme disease emergence and expansion to a number of key environmental drivers. These include habitat, climate, and wild animal host community conditions that determine environmental suitability for vectors and pathogen transmission cycles, and patterns of dispersion of ticks and pathogens by migratory birds and other animal hosts. The conditions that determine environmental suitability for *I. scapularis* are driven mostly by climate and woodland habitat. Woodland habitats provide the wild animal hosts of the tick (which is parasitic at all three feeding stages), reservoir hosts of *B. burgdorferi* (rodents and birds), as well as a litter layer that provides refuges for the ticks during periods of extreme high and low temperature and humidity [1,10]. While *I. scapularis* ticks die when exposed to freezing temperatures ($<-5\,^\circ$C), the forest litter layer protects the ticks even when air temperatures are far below zero in the Canadian winter [11]. Therefore, as long as suitable woodland habitat is present, cold winter temperatures do not limit the survival of the ticks. However, it is thought that cold temperatures limit the survival of tick populations via effects of temperature on development from one tick stage to the next. The colder the temperature, the longer the life cycle and, given constant daily per-capita mortality of ticks, the fewer the ticks that survive the whole life cycle. At a lower threshold temperature, the lifecycle is so long that the probability that a larval tick survives to be a fed adult falls below 1, and populations of the tick cannot persist [12]. The *I. scapularis* life cycle is 2 to 3 years long, and the most convenient index of temperature conditions of relevance for the survival and abundance of *I. scapularis* populations is cumulative degree-days $>0\,^\circ$C [13]. If temperature conditions and habitat were combined into a risk algorithm for the occurrence of *I. scapularis* populations and by inference in this part of North America, Lyme disease risk, this would provide key information for public health in designing surveillance programs and risk communications to the public. In the absence of a vaccine, the control of Lyme disease comprises a combination of management of the risk in the environment by reducing infected tick density or reducing contact between people and ticks, tick-bite prevention, and prompt removal of ticks [9]. Knowing which human populations in Canada are currently at risk and which populations may be at risk in the future would greatly assist Lyme disease prevention efforts.

Environmental determinants derived from Earth observation (EO) images are increasingly used in the risk modeling and mapping of infectious diseases, because they offer numerous advantages compared to field data or meteorological data [14,15]. These data provide continuous spatial and temporal coverage on a standard and regular basis: spatial, temporal, and spectral resolutions are preserved from one observation to another for the same satellite/sensor system, which are continuously in operation. They are freely accessible or at a low cost, and they cover large territories, including remote or inaccessible regions, at various spatial resolutions and temporal frequencies. They can be used to characterize past and current environmental conditions associated with the presence or abundance of disease vectors at global, regional, and local scales. The continuous development and improvement of EO programs and systems (e.g., higher spatial and temporal resolution, greater numbers, and quality of EO image derivatives), offers increasing opportunities for mapping and monitoring the environmental risk from infectious diseases. This is particularly needed for climate-sensitive infectious diseases that are specific to the ecosystem,

such as vector-borne diseases, whose emergence, maintenance/endemicity, and spread may be driven by climate and environmental changes [1,16].

In Canada, both EO data proxies and/or the ground level for these variables have been sourced to explore the ecology of Lyme disease to improve predictions [17–19], to interpret surveillance data [13,19], and ultimately to develop risk maps. However, environmental and climate data used for *I. scapularis* and Lyme disease risk modeling and mapping are often at low spatial resolutions or from sparsely distributed meteorological stations. Ref. [17] used data from the Moderate Resolution Imaging Spectroradiometer (MODIS) product MOD11B1 (monthly average surface temperatures with a spatial resolution of 5.6 km) to map the risk of Lyme disease in eastern Ontario between 2000 and 2013 [17]. EO data from the MODIS product MOD11C3 (monthly average surface temperature with a spatial resolution of 5.6 km) covering the period from 2009 to 2014 were used by Gabriele-Rivet et al. (2015) to study the geographical distribution of the tick *I. scapularis* as well as other species of Ixodid ticks in the province of New Brunswick [18]. This same product was used to map the risk of *I. scapularis* in the Canadian prairie provinces (Manitoba, Saskatchewan, Alberta) between 2009 and 2014 [20]. The Normalized Difference Vegetation Index (NDVI) derived from Landsat 7 Enhanced Thematic Mapper Plus (ETM+) images (30 m spatial resolution, 16 days temporal resolution) acquired in June 2001 was used to study the risk factors for the occurrence of *I. scapularis* in the south of the province of Quebec [13]. Several other studies used meteorological data [2,21–23] or climate models [8,12] with lower spatial resolution and coverage to model and map the risk of *I. scapularis* in Canada and North America. However, landscape heterogeneity, fragmentation of forest environments, expansion of mixed areas of woodland, agricultural, and residential surfaces, and increase of climatic variability generate microclimatic conditions and microhabitats that can provide important variation in the spatial distribution of tick populations at fine scales [15,24]. Hence, there is a need to model and map the risk of Lyme disease at finer scales in order to adapt and improve control and prevention strategies at local scales and in mixed and fragmented environments. There are no products covering Canada that can meet this need on a regular basis to support public health activities compared to EO products that have been developed to support areas such as agriculture or forestry [15]. This need requires EO data with better spatial resolution (<1 km) and greater temporal frequency (daily or weekly averages for the calculation of degree days) covering a vast territory such as Canada. This poses the challenge of processing and analyzing massive amounts of EO data [15].

Previous studies have used EO data to explore spatial variation in environmental suitability, over relatively short timescales, for vector-borne diseases [17,18,20,25–27], mostly from the perspective of ecological studies. However, here, we explore the use of EO data for public health purposes to identify how environmental suitability for the Lyme disease vector *I. scapularis* has changed over a 16-year timescale when it is thought that a warming climate has driven the range expansion of the tick and Lyme disease risk into Canada. We develop a proof of concept of *I. scapularis* risk map product in Canada based on both better spatial resolution and greater temporal frequency EO data. Specifically, we characterize the spatial and temporal variability of cumulative annual surface degree-days above 0 °C of forest habitats at a local and regional scale and map the environmental risk of *I. scapularis* in central and eastern Canada by compiling repetitive data with a spatial resolution of $500 \text{ m}^{-1}$ km.

## 2. Materials and Methods

### 2.1. Study Area

The study area was central and eastern Canada, including the provinces of Manitoba, Ontario, Quebec, New Brunswick, and Nova Scotia (Figure 1). Central and eastern Canada are the most affected by *Ixodes scapularis*-borne diseases such as Lyme disease.

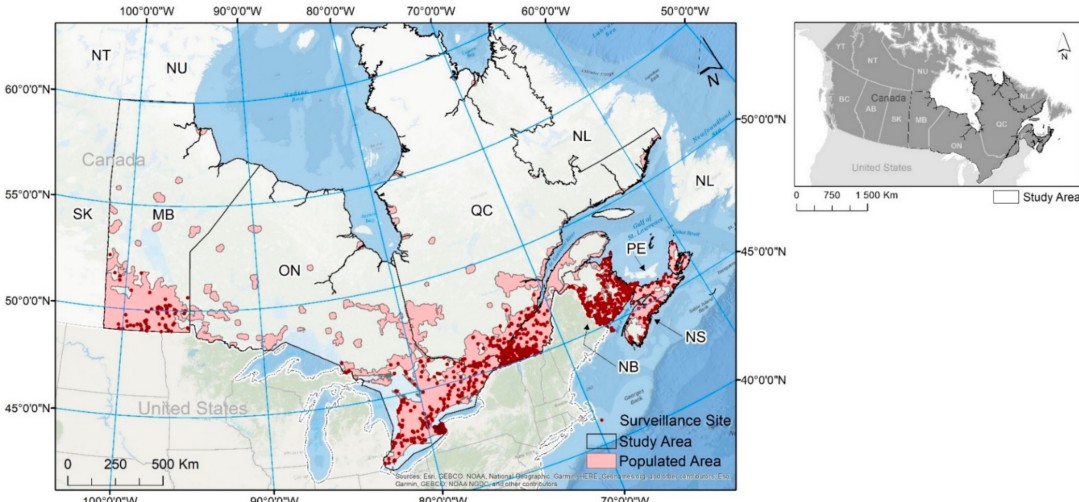

**Figure 1.** Study area showing tick surveillance sites. Abbreviations: MB, Manitoba; NB, New Brunswick; NL, Newfoundland and Labrador; NS, Nova Scotia; NT, Northwest Territory; NU, Nunavut; ON, Ontario; PEI, Prince Edward Island; QC, Quebec; SK, Saskatchewan.

### 2.2. Earth Observation Data Processing

EO images from the Moderate Resolution Imaging Spectroradiometer (MODIS) on-board the Terra satellite were used. The images include the Terra MODIS Land Surface Temperature and Emissivity product (MOD11A2), Version 6 (Wan 2017), and the Terra MODIS Land Cover Type product (MCD12Q1), Version 6 (Sulla-Menashe and Friedl 2018). The MOD11A2 product is an 8 days-per-pixel averaging land surface temperature (LST) with a 1 km spatial resolution. The MCD12Q1 product is a 500 meters' spatial resolution land cover (LC) image produced annually. The MODIS images used in this study were acquired from 2000 to 2015. Images were downloaded from the National Aeronautics and Space Administration (NASA) Earth science data platform Earthdata ( https://earthdata.nasa.gov/). A total of 6624 files were downloaded for LST images and 135 files were downloaded for LC images to cover the study area and time period.

EO data processing included format conversion, reprojection, mosaicking, the creation of masks for pixel selection, image resampling, and integration. MODIS images were converted from their original Hierarchical Data Format file to PCI Geomatics Database File (PIX) format. They were reprojected from MODIS Sinusoidal projection to Lambert conformal conic projection. LST images from the MOD11A2 product were resampled from 1 km to 500 m spatial resolution as land cover (LC) images from the MCD12Q1 product using the nearest-neighbor resampling method [28]. Datasets of LST images and LC images were merged in same database files according to their date and coverage area. All processed data were mosaicked on the study area per variable and date. Using quality assessment values related to each MODIS product, bitmap masks were created to identify cloud, cloud shadow, adjacent to cloud, cirrus, snow, ice, water, and high LST error (>2 K) pixels and to select valid pixels for LST and LC. Valid pixels selected for LST estimation exclude cloud, cloud shadow, adjacent to cloud, cirrus, and high LST error pixels. The MODIS LST scale factor was applied to derive LST in Kelvin unit (K). LST Day and LST Night channels were used to estimate a day/night average LST ($LST_{dnm}$, Equation (1)).

$$LST_{dnm} = (LST\ Day + LST\ Night)/2. \tag{1}$$

LC was derived using the Land Cover Type 1 layer of the MCD12Q1 product. This layer is based on the Annual International Geosphere-Biosphere Programme classification. It has 16 LC classes including Evergreen Needleleaf Forests (ENF), Evergreen Broadleaf Forests (EBF), Deciduous Needleleaf Forests (DNF), Deciduous Broadleaf Forests (DBF), Mixed Forests (MXF), Closed Shrublands (CSL), Open Shrublands (OSL), Woody Savannas,

Savannas, Grasslands, Permanent Wetlands, Croplands, Urban and Built-up Lands, Cropland/Natural Vegetation Mosaics, Permanent Snow and Ice, Barren, and Water Bodies. Woodland land cover classes (ENF, EBF, DNF, DBF, MXF, CSL and OSL) were combined to map suitable vector habitat of Forest (F = 1), while all other habitats were combined to map unsuitable habitat (F = 0).

### 2.3. Estimation of Cumulative Annual Surface Degree Days

Surface degree days were estimated as the daily average LST values above 0 °C (SDD > 0 °C) (e.g., if on one day, the daily average LST = −2.3 °C (i.e., <= 0 °C), then SDD = 0; but if the daily average LST on that day is = 2.3 °C (>0 °C), then SDD = 2.3 °C). Cumulative annual surface degree days >0 °C (henceforth abbreviated to CASDD) were calculated by summing SDD >0 °C values over the year.

CASDDs were validated using meteorological station data. As in no-stress conditions, the surface temperature of woodlands with full canopy is close to the near-surface air temperature [14,29,30], CASDDs of forest cover (CSDFs) were compared to cumulative annual degree days >0 °C (CADD) derived from air temperature observed by meteorological stations located in areas with woodlands as the dominant vegetation. 2015 was used as the validation year. Air temperature data from 235 meteorological stations of the Environment and Climate Change Canada network were downloaded from Climatedata.ca. Woodland land cover classes from MODIS Land Cover product (MCD12Q1) and percent vegetation cover (PVC) derived from MODIS Vegetation Continuous Fields product (MOD44B) were used to identify forest covers and dominant vegetation areas. Percent tree cover and percent non-tree vegetation cover from the MOD44B product [31,32] were summed to estimate PVC. Final selection of the meteorological stations used for the validation assessment was based on the following criteria: (1) stations with less than one hundred (100) days of missing data in the year or with less than ten (10) days missing data in a summer month (higher values of degree days >0 °C in the year); (2) stations that were located in a 3 km buffer area with an average PVC equal or greater than 85% and covered by more than fifty percent (50%) of forest. Missing values of daily degree days > 0 °C of each selected meteorological station were filled using the nearest date. Average values of CSDF in the 3 km buffer around meteorological stations were used for the comparison with CADD.

A simple linear regression model was used to assess the relationship between CSDF and CADD. Root Mean Square Error (RMSE, Equation (2) and Mean Absolute Error (MAE, Equation (3) were used to quantify the accuracy of CSDF in reference to CADD.

$$\text{RMSE} = \sqrt{\frac{\sum_{i=1}^{N}\left(\text{CSDF}_p - \text{CSDF}_o\right)^2}{N}} \tag{2}$$

$$\text{MAE} = \frac{\sum_{i=1}^{N}\left|\text{CSDF}_p - \text{CSDF}_o\right|}{N} \tag{3}$$

N is the number of meteorological stations used for the validation. $\text{CSDF}_p$ and $\text{CSDF}_o$ are the CSDF predicted by the linear regression model and the CSDF derived from EO images, respectively.

The slope of the temporal linear relationship between CSDF and year at each pixel was used to estimate the rate of increase or decrease of CSDF between 2000 and 2015 [17,33,34].

### 2.4. Risk Mapping

Two types of risk maps were produced. First, a simple multiplication of CASDD values and the forest classification produced a map of cumulative annual surface degree-days above 0 °C of forest habitats (CSDF). Second, a risk map of an index of *I. scapularis* tick density was obtained as follows. CASDD values for *I. scapularis* risk maps for 2000 to 2015 used a relationship between CADD and *I. scapularis* density, which was obtained from analyses conducted using an *I. scapularis* population model [21] and subsequently

validated by field studies [18]. Combining this relationship with the presence/absence of forest produced an index for the risk of *I. scapularis* (RIS) (Equation (4)).

$$\text{RIS} = \text{F} \times ((0.436 \times \text{CADD}) - 1232), \tag{4}$$

where

RIS = Risk of *I. scapularis*
F = Binary variable related to the vector habitat. F = 1 if forest, F = 0 if not forest
CADD = Cumulative annual degree days >0 °C

The difference in these two risk maps is that using RIS, there is a lower cut-off threshold for CADD (circa 2800 DD > 0 °C) below which *I. scapularis* populations cannot become established. This limits the geographic scope of tick population establishment and has been found to be robust in surveillance [2,18].

The numbers of successive years with RIS > 0 was calculated and mapped to assess for each elementary surface (pixel of 500m$^2$) of forest whether favorable temperature conditions for *I. scapularis* ticks are maintained there long enough to facilitate their development and reproduction.

Spatio-temporal variations in CSDF and RIS were analyzed at the provincial scale, the scale of populated areas in each province, and at the scale of surveillance sites in each province (Figure 1). Populated areas of each province correspond to the population ecumene, which was defined as inhabited lands where most people have made their permanent home [35]. The population Ecumene Census Division cartographic boundary file from Statistics Canada [35] was used to identify populated areas in order to assess the degree of intersection of environmental suitability and human populations, which is an important consideration in public health responses to Lyme disease [36].

Algorithms developed in EASI programming language, Geomatica 2018 software (PCI Geomatics, Markham, ON, USA), were used to process EO images, calculate CSDF and RIS, and analyze their spatio-temporal variations. ArcGis Desktop 10.7 (Environmental Systems Research Institute, Redlands, CA, USA) was used to produce the risk maps.

### 2.5. Validation of Risk Maps

The performance of the risk maps was tested by comparisons with field surveillance for *I. scapularis* ticks against data on temperature conditions and woodland types obtained from the EO data in logistic regression, and against data on temperature only by Receiver Operating Characteristic (ROC) analysis.

### 2.5.1. Surveillance Data

Data for validation were from surveillance activities conducted from 2007 to 2013 in the Canadian provinces of Manitoba, Ontario, Quebec, New Brunswick, and Nova Scotia [37,38], which were supplemented with more recent data from surveillance and specific studies from 2014 to 2018 in Ontario [39] and New Brunswick and Quebec [40]. Surveillance data were obtained by drag-sampling for questing ticks as previously described [41], and in most cases, the presence of one *I. scapularis* tick was considered as evidence of established reproducing populations of the tick [41]. Some sites were visited in multiple years, and if these sites were negative and then positive for ticks in a subsequent year, they were considered positive. However, if sites were positive in one year and then negative in subsequent years, they were considered negative.

### 2.5.2. Explanatory Variables

The explanatory variables were CSDF and percentage of land cover of different types of woodlands (evergreen needle forest = ENF, deciduous broadleaf forest = DBF, and mixed forest = MXT). The CSDF for each study site was obtained as the mean value for pixels within a buffer of 2 km from the site for each year for which MODIS data were available from 2000 to 2015. The value used in statistical analysis was the maximum value for

CSDF (DDMax) for the years for which data were available for each site. This was chosen over a mean DD > 0 °C value because there were missing data for some years, and the missing years differed amongst the sites. The percentage of land cover of different types woodlands was calculated for a buffer of 2 km around each site. The province of study was considered as a categorical variable in analysis of the complete dataset, as the data from different provinces may vary in terms of the types of site selected for surveillance or study, and because woodland types within the broad woodland categories vary geographically in Canada.

2.5.3. Statistical Analysis

First, the existence of expected, biologically plausible associations between the explanatory variables and the presence or absence of *I. scapularis* at study sites (i.e., with warmer temperature conditions and deciduous woodlands) were explored in logistic regression models in Stata SE version 11 (Statacorp, College Station, TX, USA). As a first step, data from all provinces were analyzed together. Explanatory variables were the different forest types and DDMax for each site that were tested in bivariable analyses. The assumption of linearity between explanatory variables and log odds of the outcome was explored by visual inspection of Lowess smoothed graphs produced in STATA, and polynomial terms were developed as needed. Variables significant at the $p < 0.1$ level were included in a multivariable model, and a final model contained only variables significant at the $p < 0.05$ level. These analyses were repeated using temperature and woodland type data for data from each province.

Second, the ability of DDMax to predict positive and negative sites was explored by ROC analysis in Stata to assess the overall predictive power by the area under the ROC curve (AUC). Values for the sensitivity and specificity of detection were obtained for a DDMax of 2800, which is the model-derived minimum value of temperature condition for the survival of *I. scapularis* populations [21] found to be appropriate in studies of surveillance for the tick [18,22]. AUC values were also obtained for data for each province individually.

## 3. Results

*3.1. Validation of Cumulative Annual Surface Degree Days >0 °C of Forest Covers (CSDF) with Ground Station Meteorological Data*

From the 235 meteorological stations on the study area, 63 were rejected for missing data, while 52 located in dominant vegetation areas, with an average of percent vegetation cover (PVC) higher than 0.85 and a percent forest cover higher than 50% in the 3 km buffer around meteorological station, were used for the validation. Figure 2 presents the map of PVC with the meteorological stations. Cumulative annual surface degree days >0 °C of forest covered areas (CSDF) derived from MODIS images were strongly associated with cumulative annual degree days >0 °C (CADD) derived from air temperature observed by meteorological stations in dominant vegetation areas (N = 52, $R^2 > 0.86$, $p < 0.00001$). The relationship between CSDF and CADD is more described in Figure 3 and Table 1. RMSE and MAE of the CSDF in reference to CADD are 152.996 CADD > 0 °C and 127.469 CADD > 0 °C, respectively.

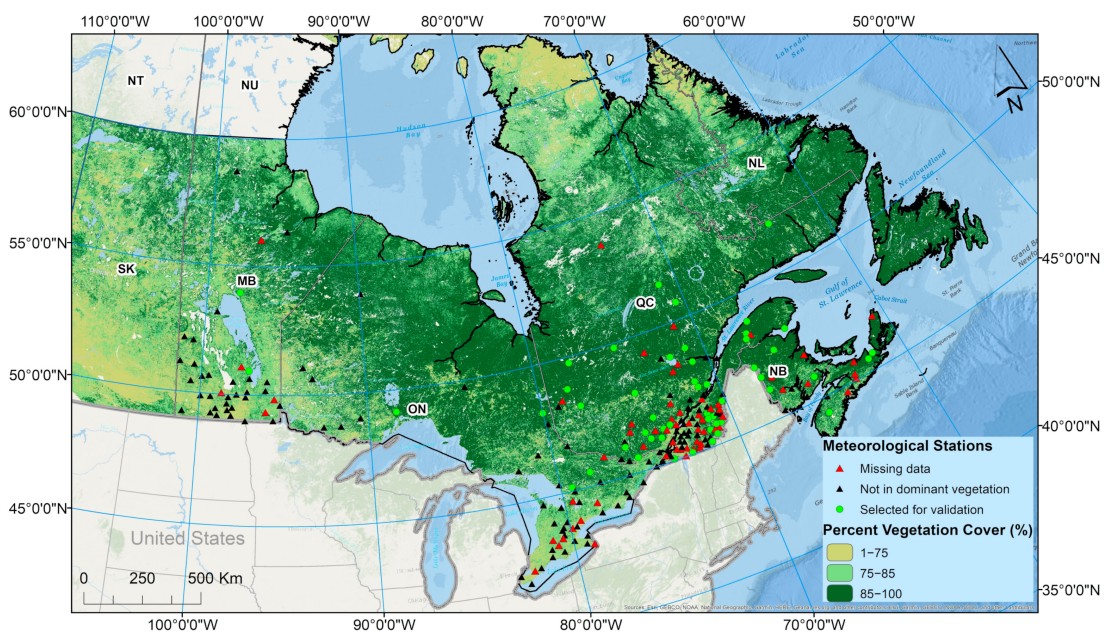

**Figure 2.** Meteorological stations used for validation were located in a 3 km buffer area with an average percent vegetation cover greater than 85% and covered more than fifty percent (50%) by forest.

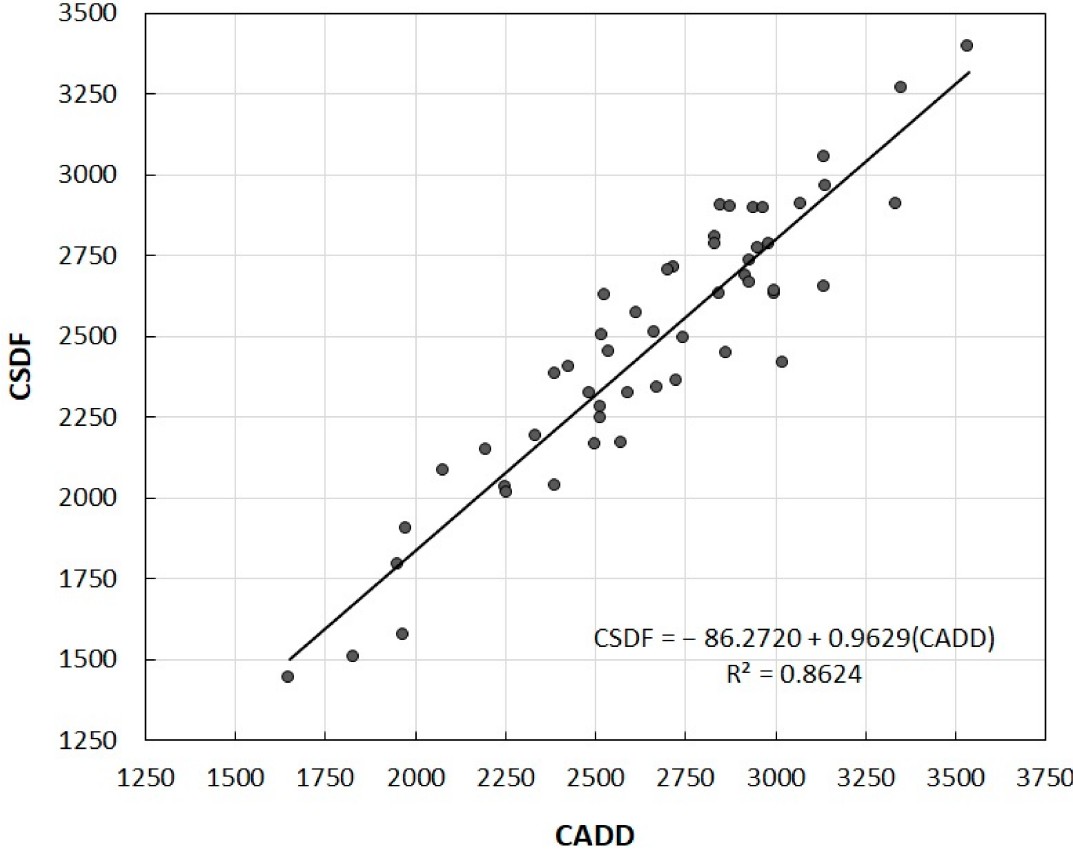

$$CSDF = -86.2720 + 0.9629(CADD)$$
$$R^2 = 0.8624$$

**Figure 3.** Relationship between cumulative annual surface degree days >0 °C of forest covered areas (CSDF) derived from MODIS images and cumulative annual degree days >0 °C (CADD) derived from air temperature observed by meteorological stations in dominant vegetation areas.

**Table 1.** Coefficients of the linear relationship between cumulative annual surface degree days >0 °C of forest covers (CSDF) derived from Moderate Resolution Imaging Spectroradiometer (MODIS) images and cumulative annual degree days >0 °C (CADD) derived from air temperature observed by meteorological stations in dominant vegetation areas.

|  | Estimate | Std. Error | t Value | Pr (> \| t \|) |
|---|---|---|---|---|
| **(Intercept)** | −86.272 | 146.6681 | −0.588 | 0.559 |
| **CADD** | 0.96287 | 0.05438 | 17.706 | $<2 \times 10^{-16}$ *** |

*** Residual standard error: 156 on 50 degrees of freedom; Multiple R2: 0.8624, Adjusted R2: 0.8597; F-statistic: 313.5 on 1 and 50 DF, *p*-value: $< 2 \times 10{-}16$; Root Mean Square Error: 152.996 CADD > 0 °C; Mean Absolute Error: 127.469 CADD > 0 °C.

*3.2. Spatial and Temporal Variations of Cumulative Annual Surface Degree-Days above 0 °C of Forest Habitats*

CSDF varied up to a maximum of 5309 DD > 0 °C in space and time (Figure 4). The average of CSDF for the entire study region varied between 1691 and 2222 DD > 0 °C from 2000 to 2015, with maximum values between 4203 and 5309 DD > 0 °C. The CSDF increased in the study region between 2000 and 2015. The lowest values of CSDF were observed in 2000 and the highest values were observed in 2012 (Figure 4).

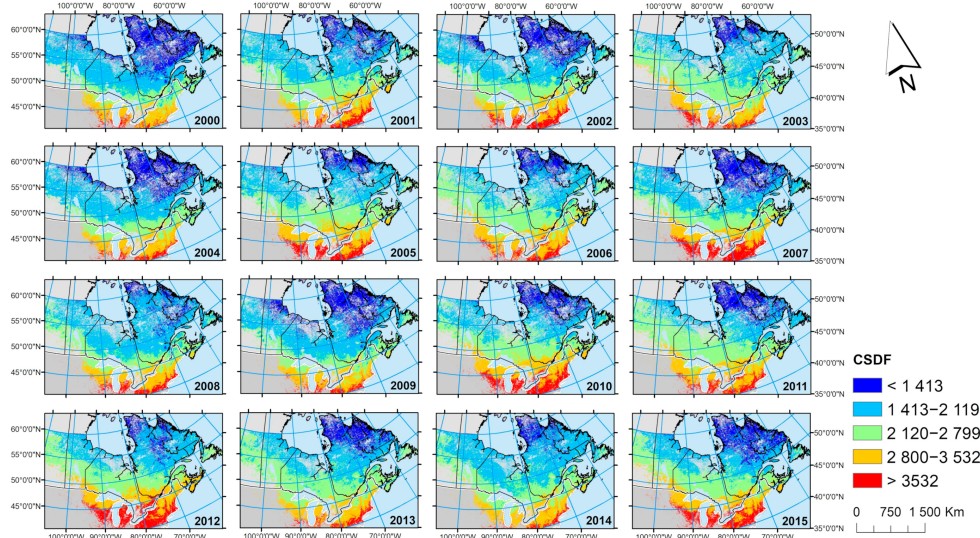

**Figure 4.** Annual cumulative surface degree days > 0 °C of forest habitats (CSDF) in Central and Eastern Canada from 2000 to 2015.

The average CSDF for 2000 to 2015 (Figure 5) is characterized by a temperature gradient from south to north, with higher temperatures in the south of the provinces of Manitoba, Ontario, Quebec, and Nova Scotia (Figure 5). The highest values were in southern Ontario and Quebec. The CSDF decreases toward the north to reach the geographical limit of the minimum threshold for the reproduction of tick populations (2800 DD > 0 °C). Southern Ontario is the area of the study region where CSDF values remained higher throughout the study period.

Figure 6 shows the variation in the mean and maximum values of the CSDF between 2000 and 2015 at provincial, populated area, and surveillance site scales. At the provincial scale, the CSDF averages for Nova Scotia and New Brunswick were the highest and lowest for Quebec and Manitoba. However, maximum CSDF values were highest in Ontario and Quebec.

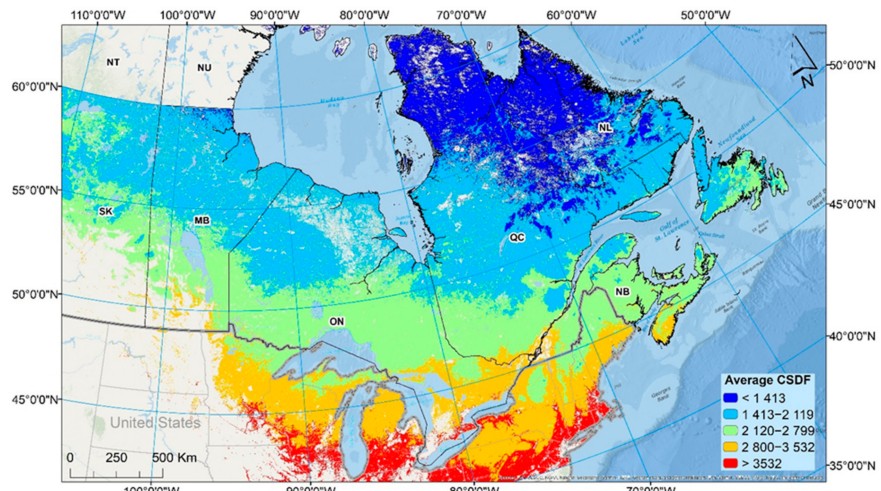

**Figure 5.** Cumulative average annual surface degree days >0 °C of forest habitats (CASDD) in Central and Eastern Canada from 2000 to 2015.

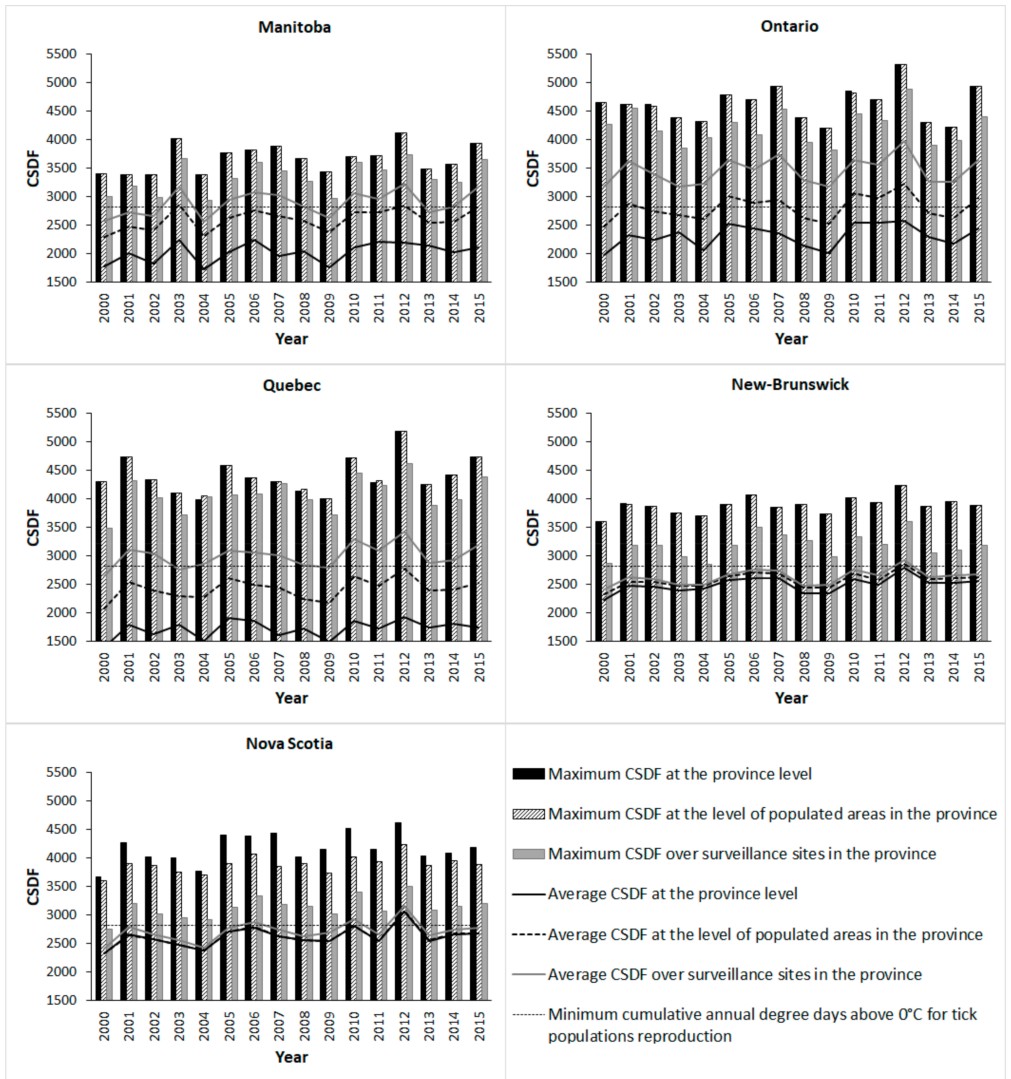

**Figure 6.** Variation of cumulative annual surface degree days above >0 °C of forest covered areas (CSDF) from 2000 to 2015 at provincial, populated area, and tick surveillance site scales.

The ranges from average to maximum CSDF was, as expected, highest in the larger provinces (Manitoba, Ontario, and Quebec), and in these provinces, the average values for surveillance sites and populated areas (which are in the south of the provinces) were higher than values for the whole province (Figure 6).

CSDF increased over time in almost all of the study region (Figure 7). This increase was more rapid in the north (>35 °C/year) where CSDFs are not yet favorable to the reproduction of tick populations. In areas where the CSDF is favorable for reproducing tick populations, the slope varied between 15 and 35 °C/year, with some exceptions in southern Manitoba, Ontario, and Quebec where it was higher than 35 °C/year. However, negative slope values were observed in several areas, including northeastern Manitoba, central and northeastern Ontario, and the extreme northwestern Quebec. These negative slope values correspond to decreases of between 1 and 10 °C/year.

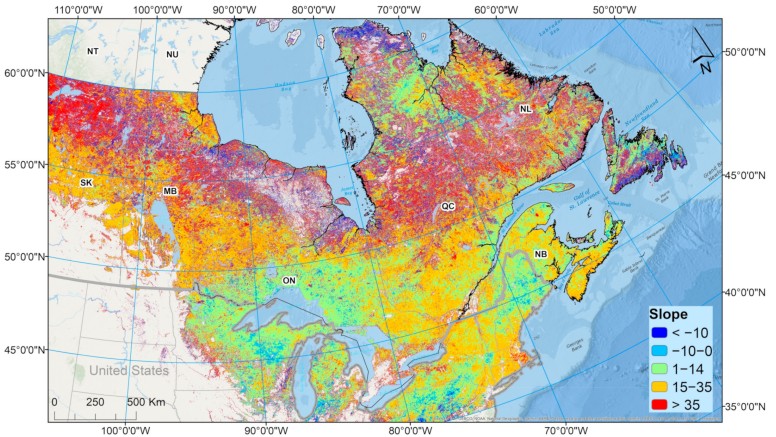

**Figure 7.** Changes in annual cumulative surface degree days >0 °C of forest covered areas in Central and Eastern Canada from 2000 to 2015.

### 3.3. Risk of Ixodes Scapularis

The risk of I. scapularis (RIS) is shown for each year, and as an average for the period, in Figures 8 and 9 respectively. As for CSDF, the highest RIS values were observed for Ontario and Quebec, while the RIS values were lowest on average in New Brunswick. In populated areas, the mean RIS values were higher than provincial averages, and they were higher still for surveillance sites (Figure 10).

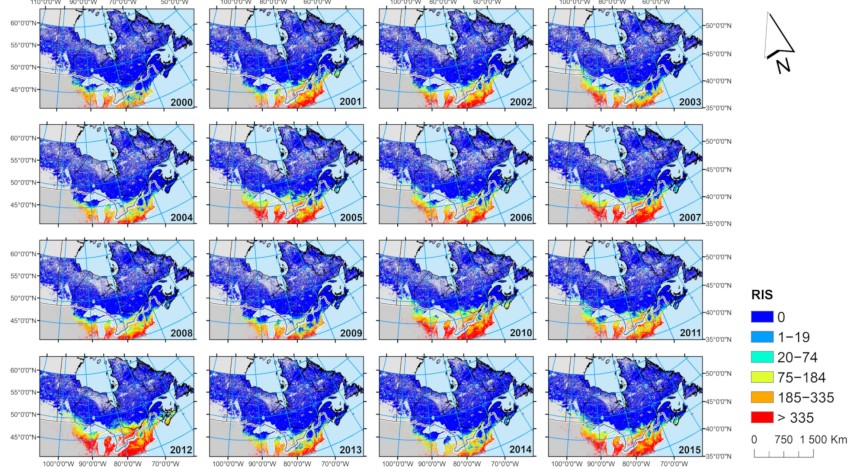

**Figure 8.** Maps of risk of *I. scapularis* (RIS) in Central and Eastern Canada from 2000 to 2015.

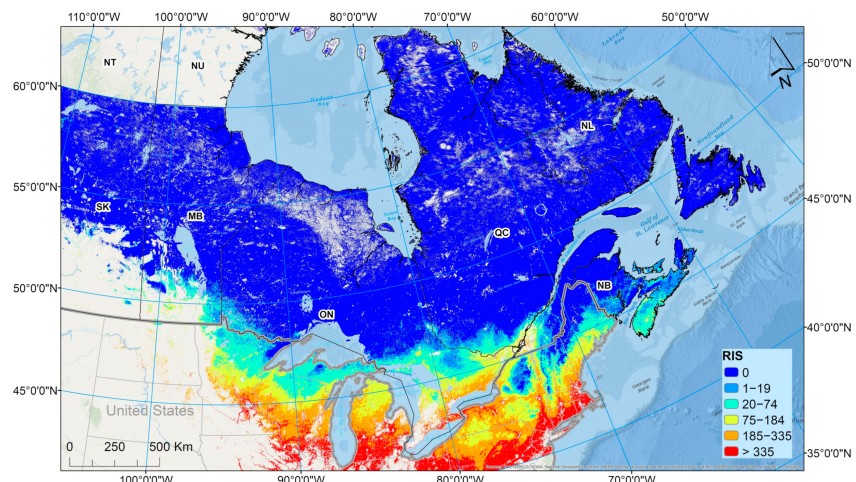

**Figure 9.** Average risk of *Ixodes scapularis* (RIS) in Central and Eastern Canada from 2000 to 2015.

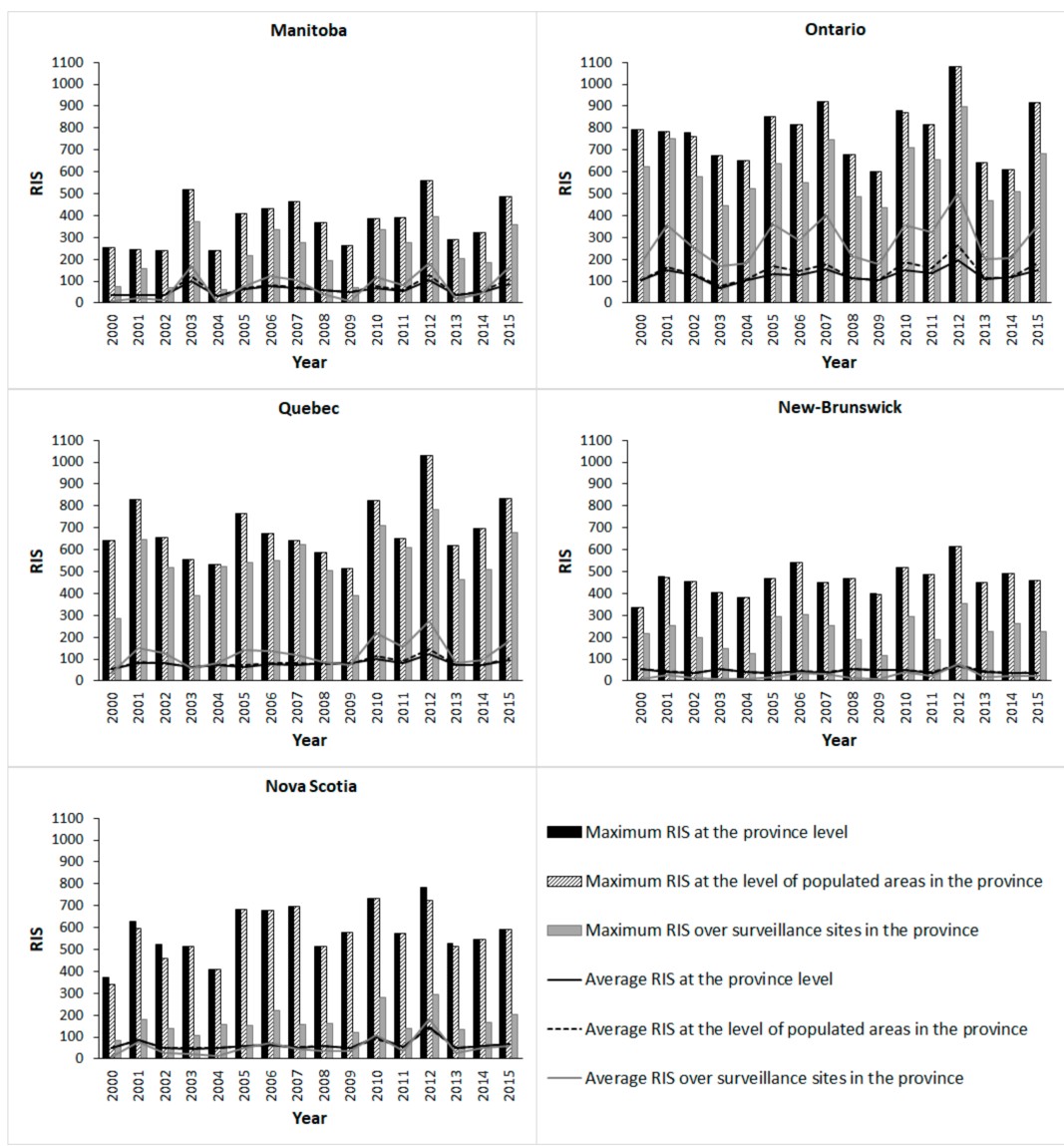

**Figure 10.** Variation of risk of *Ixodes scapularis* (RIS) from 2000 to 2015 at province, populated area, and tick surveillance site scales.

The proportion of the populated areas with a positive RIS varied between 6% and 47% (average 22%) in the period from 2000 to 2012 (Figure 10). This proportion was on average higher in Ontario (37%) and Nova Scotia (26%), and lower in Manitoba (11%), Quebec (14%) and New Brunswick (14%).

Overall, the numbers of successive years with RIS > 0 was highest in the southern regions (Figure 11).

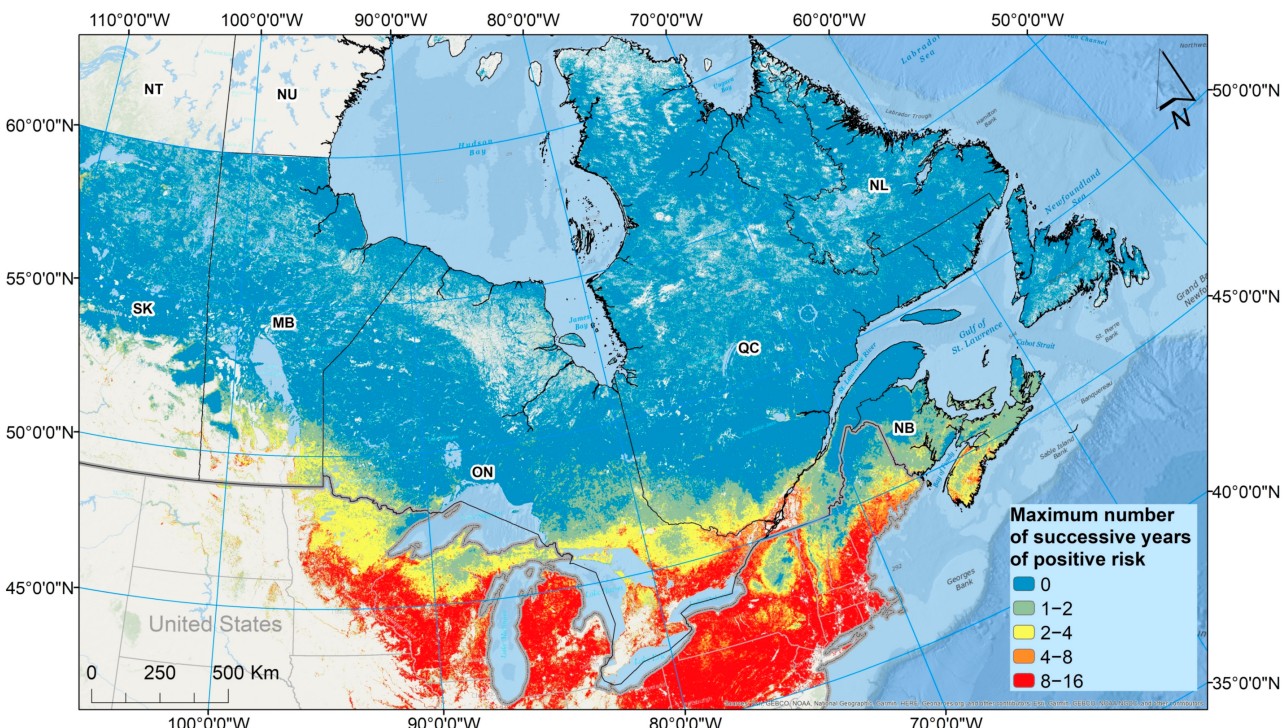

**Figure 11.** Maximum successive years of positive risk of *Ixodes scapularis* (RIS) in Central and Eastern Canada from 2000 to 2015.

### 3.4. Surveillance Data

Data from 1159 sites (of which 285 were positive for *I. scapularis*) were used in analysis after the removal of duplicates and a few sites for which explanatory data were incomplete. These data comprised 57 sites from Manitoba (22 were positive), 372 in Ontario (75 were positive), 399 in Quebec (151 were positive), 275 in New Brunswick (27 were positive), and 49 sites in Nova Scotia (9 were positive). Due to the small number of sites in Nova Scotia and their geographic proximity, data from New Brunswick and Nova Scotia were combined for analysis as the "Maritimes".

### 3.5. Logistic Regression Analysis

Following examination of Lowess curves, a quadratic term for DDMax was used in analyses using data from all provinces combined, and those using data from Ontario and Quebec. In all analyses, DDMax and DDMax$^2$ (when used) were significantly associated with the likelihood of presence of *I. scapularis*. The proportion of land cover that was broadleaf forest (DBF) were significantly associated with the likelihood of presence of *I. scapularis* for Ontario and Quebec. When data from all provinces were analyzed together, there were significant differences amongst provinces (Table 2).

**Table 2.** Outcome of logistic regression models in which the outcome was presence/absence of *Ixodes scapularis* and temperature variables (maximum annual cumulative number of degree-days >0 °C ([DDMax and a quadratic term DDMax2]), the proportion of land cover that was broadleaf forest (DBF) and province (in the model using data from all provinces) were explanatory variables.

| Explanatory Variable | Odds Ratio (95% Cis) | Wald z | *p* |
|---|---|---|---|
| | All provinces | | |
| DDMax | 1.033 (1.025–1.041) | 8.41 | <0.001 |
| DDMax2 | 0.999 (0.999–0.999) | −8.24 | <0.001 |
| DBF | 1.028 (1.014–1.42) | 3.95 | <0.001 |
| Province | | | |
| Manitoba = reference | | | |
| Ontario | 0.257 (0.126–0.524) | −3.74 | <0.001 |
| Quebec | 0.567 (0.296–1.085) | −1.71 | <0.1 |
| New Brunswick | 0.743 (0.339–1.625) | −0.74 | > 0.1 |
| Nova Scotia | 0.420 (0.163–1.1.085) | −1.79 | <0.1 |
| | Manitoba | | |
| DDMax | 1.003 (1.000–1.005) | 2.13 | <0.05 |
| | Ontario | | |
| DDMax | 1.058 (1.031–1.086) | 4.29 | <0.001 |
| DDMax2 | 0.999 (0.999–0.999) | −4.34 | <0.001 |
| DBF | 1.087 (1.039–1.139) | 3.61 | <0.001 |
| | Quebec | | |
| DDMax | 1.038 (1.023–1.053) | 5.08 | <0.001 |
| DDMax2 | 0.999 (0.999–0.999) | 14.87 | <0.001 |
| DBF | 1.022 (1.007–1.037) | 2.94 | <0.01 |
| | Maritimes | | |
| DDMax | 1.004 (1.002–1.006) | 3.76 | <0.001 |

*3.6. ROC Analysis*

AUC values were moderate in all cases (70% or less) and particularly low for Manitoba and Ontario (Table 3). However, the sensitivity of detection of *I. scapularis*-positive sites with a cut-off of 2800 degree-days >0 °C was very high being at, or approaching, 100%. High sensitivity values (>80%) were maintained to a DDMax value of 3300, above which sensitivity declined rapidly, so low AUC values were due to low specificity, which increased linearly across the range of DDMax values (Figure 12). As an illustration, with a cut-off of 3000 degree-days >0 °C, there was a small number (13/231) of possibly "false positive" sites (i.e., sites with degree-days >0 °C of <3000, but with one or more ticks), most positive sites (271) had degree-days >0 °C of < 3000, but approximately two-thirds of sites with degree-days >0 °C of <3000 (649 sites) had no ticks and could be seen as "false negative" sites.

**Table 3.** Results of Receiver Operating Characteristic (ROC) analysis. Area under the ROC curve (AUC) values are shown alongside values for sensitivity and specificity of discrimination of *Ixodes scapularis* positive and negative with a maximum value for CSDF (DDMax) cut-off of 2800 degree-days >0 °C.

| Data | ROC AUC (SE) | Sensitivity of Detecting *I. Scapularis* if DDMax > 2800 | Specificity of Detecting *I. Scapularis* if DDMax > 2800 |
|---|---|---|---|
| All provinces | 0.604 (0.016) | 99.3% | 5.31% |
| Manitoba | 0.521 (0.053) | 100% | 5.88% |
| Ontario | 0.401 (0.032) | 100% | 0.34% |
| Quebec | 0.650 (0.027) | 100% | 5.65% |
| Maritimes | 0.700 (0.049) | 94.4% | 9.80% |

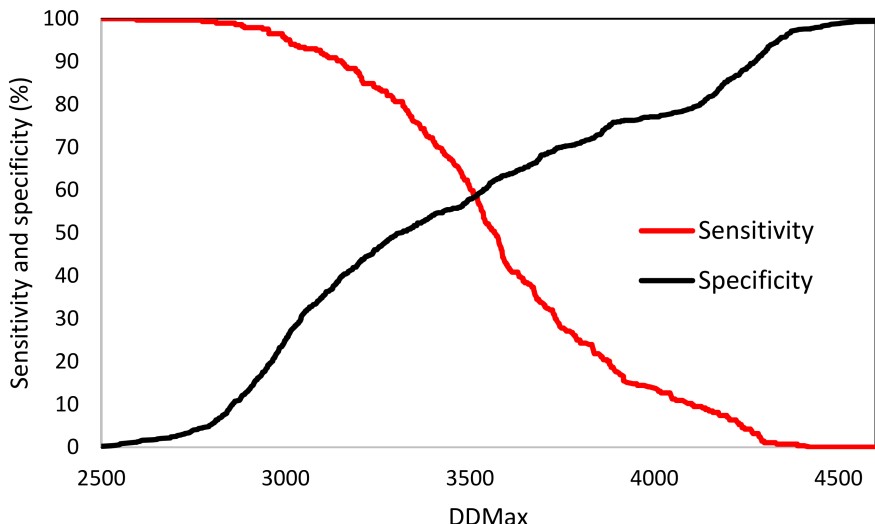

**Figure 12.** Variation in specificity and sensitivity of detection of *Ixodes scapularis* for different values of the maximum cumulative annual surface degree-days above 0 °C within forest covered areas (DDMax).

## 4. Discussion

In this study, we have developed risk maps for the Lyme disease vector *I. scapularis* in central and eastern Canada using EO data on temperature and land use. Infectious diseases such as Lyme disease, whether on a global, regional, or local level, are public health concerns that affect individuals but pose escalating threats to the larger population. Since generating evidence-based knowledge about these diseases is key to managing them and reducing their impact, the capacity of EO satellites to quickly and accurately collect extensive datasets on the changing drivers for disease occurrence and spread gives public health a tactical advantage in predicting disease risks.

The maps performed as well as could be expected in validation against surveillance data. Specificity was relatively low, i.e., the maps tended to suggest that more sites should have tick populations in surveillance sites than what was observed. This was expected for two reasons. First, *I. scapularis* is expanding its range into Canada in the regions where surveillance occurs, and not all woodlands at the expanding northern edge to the range acquire detectable tick populations at the same time, even though they may have suitable habitat and climate (reviewed in [40]). So, not all suitable sites would be expected to have tick populations at the time surveillance took place, as shown by surveillance studies [39]. Second, while climate may be suitable, and woodlands present, not all woodlands may be suitable for *I. scapularis* populations (see below). However, from a public health perspective, it is preferable that tools such as risk maps, which help decision-making, have a more precautionary approach that will include most of the true positive sites and some false positive sites. By having high sensitivity and lower specificity, these risk maps do take a precautionary approach. It should be noted that at some surveillance sites, particularly in Manitoba, low numbers of adult ticks were collected by drag sampling, but immature ticks have not subsequently been found on drags or by the capture of small mammal hosts (personal communication Dr. K. Rochon). This suggests that at some sites, ticks collected were actually adventitious ticks rather than coming from reproducing populations at the surveillance site [13], so the maps may be more sensitive in detecting reproducing tick populations than estimated here. In the future, more systematic surveillance for ticks that includes re-visiting of sites year-on-year will allow us to better calibrate rates of invasion of ticks into suitable regions and better evaluate the performance of risk maps.

The advantages of using EO data over ground-level observations are well illustrated by the risk maps. The maps have the same spatial resolution from the more heavily populated south of Canada (where climate station density is high) to the sparsely populated north

where terrestrial climate stations are very sparse. However, risk information is equally important to populations in the north and the south. The maps illuminated an important issue regarding spatio-temporal variations in temperature conditions both over the short and long term, which again can only be fully explored by EO data that have constant resolution in space and time. There was evidence of overall increasing temperatures over the time scale of the data used here (Figure 7) that is consistent with the observed climate warming in Canada over a longer timescale that has been associated with range expansion of the tick into and through Canada [2,6]. There was evidence that warming was greater and faster the further north, which is again consistent with terrestrial observations and climate model-based projections of how the climate is expected to change now and in the future [16]. However, during the study period here, there was considerable inter-annual variation in cumulative DD > 0 °C with temperatures highest in 2012 and lowest in 2000 (which is consistent with terrestrial observations [42]). The life cycle of *I. scapularis* is multi-year, so increased temperature above a threshold of 2800 DD > 0 °C will not allow a tick population to become established. Multiple years with DD > 0 °C above the threshold would be required for the ticks to become established and then for *B. burgdorferi* transmission cycles to become efficient [23]. Therefore, mapping where temperature conditions are above a threshold for a number of consecutive years (Figure 11) may be the most useful product to drive the design of programs of field surveillance for *I. scapularis*, while the map of the average RIS value over multiple years (Figure 9) may be the most useful for communicating the risk of Lyme disease to the public.

Of importance for Lyme disease risk in Canada is the observation that the most climatically suitable parts of Canada are the southernmost regions where the most populous areas are also found, and suitable forest habitat occurs here, too. Therefore, small changes in tick distributions and tick abundance in the coming years could have large impacts on the incidence of Lyme disease. The risk maps provide guidance on where surveillance for emerging tick populations may be most usefully directed in these regions.

The risk maps illustrated some inter-provincial differences in environmental suitability that are reflected in surveillance data. On average, surveillance sites in the Maritime provinces were located in areas where temperature conditions were often below the threshold for tick population establishment, and tick populations have a patchier occurrence in these provinces compared to the warmer regions of southern Ontario and Quebec [37]. However, clearly, environmentally suitable areas exist in these provinces, and the risk maps can support the development of targeted surveillance to these areas.

The index of risk for *I. scapularis* (RIS) is more useful that CSDF in having a more clear limit for environmental suitability for *I. scapularis*. However, RIS should be considered an index only, as it is based on the numbers of adult ticks predicted in a model. Above the threshold for tick population survival, the risk of Lyme disease is likely to increase with temperature, but there are many ecological complexities. After tick populations become established, tick density will increase year-on-year until some kind of equilibrium is reached (although continuing warming may result in increasing tick density when tick populations are at equilibrium). At the same time, the prevalence of infection in ticks will increase as *B. burgdorferi* transmission cycles become established and more efficient [43]. The simple presence/absence of forest as an indicator of environmental suitability is a great simplification of how the qualities of different woodland types may determine Lyme disease risk via effects on wild animal host ranges and densities, and litter layer qualities that in turn determine tick densities and *B. burgdorferi* infection prevalence. This was evident in the validation analyses. Evaluating these qualities by field study is a work in progress, and the outcomes of these studies will allow more granular identification of Lyme disease risk habitats according to woodland types, which will likely also be identifiable using EO data [25].

Nevertheless, the increase in RIS between 2000 and 2015 is consistent with the evolution of Lyme disease risk in Canada, as elucidated by active and passive surveillance for

ticks, and surveillance for human Lyme disease cases [19,24,38,39]. Geographic variations in RIS are also consistent with observations [40,44].

Here, we used data from 2000 to 2015 to demonstrate the utility of EO data to identify environmental suitability for *I. scapularis* and thus Lyme disease risk. The method can now be implemented in concert with ongoing surveillance to review how Lyme disease risk may have increased in geographic scope year-on-year, and how risk may evolve in coming decades with projected climate change [8]. The method we propose makes it possible to use EO images to assess and map the risk of *I. scapularis* with useful accuracy and spatial resolution, which offers monitoring of the spatial variability of Lyme disease risk at local scales. This would support control and prevention activities by those responsible at municipality and health region scales. However, the method allows monitoring of changes to the extent of the risk over wider spatial scales and over longer time scales that are relevant for studying possible effects of climate change on changes in vector distributions. Indeed, the main importance of this study is the potential for application of EO data directly to produce risk information for public health to support decision making and risk communication for a vector-borne disease that is emerging due to climate change. Therefore, it may be a paradigm study to support the application of EO data to public health. However, there are a number of additional contributions to the field of application of remote sensing in spatial epidemiology. These include combining a greater temporal frequency (average of 8 days), a finer spatial resolution (500 m$^{-1}$ km), and observation over a longer period of time (16 years) compared to previous studies [17,18,20,39]. We used a new approach to estimate CSDF that offers good accuracy as evidenced by the validation against meteorological data (Table 1).

## 5. Conclusions

*Ixodes scapularis* continues its geographic range expansion into south central and southeastern Canada. This is accompanied by continued increases incidence of human Lyme disease cases. Climate change, with increasing temperatures, is thought to be facilitating this range expansion. Here, we developed risk maps for the tick vector of Lyme disease, which could be used as an indicator of Lyme disease risk. The risk maps identified spatio-temporal variations in risk that are consistent with our knowledge of the changing climate in Canada, how that is varying geographically, and how that may be impacting spatio-temporal changes in the geographic range of *I. scapularis* and Lyme disease risk. These maps can support public health objectives. They can identify risk to support communications to the public in regions where it is already known that the tick is becoming established. They can guide surveillance in possible areas of risk emergence, i.e., areas identified as environmentally suitable that are not currently known to harbor tick populations. They demonstrate key qualities of Earth observation data for risk mapping purposes in having the same resolution and accuracy across the entire geographic scope of central and eastern Canada, including regions where terrestrial observations are scarce. Furthermore, the ability of the maps to be updated annually means that effects of changes in climate on Lyme disease risk can be actively monitored. There is also a need to explore in more detail the effect of different woodland types on *I. scapularis* survival and Lyme disease risk, and to associate these with Earth Observation proxies for the different habitat types. Finally, further studies are needed to continue improving this Earth observation capacity and to expand public health applications of Earth observation technology to meet the information needs for a broader range of health risks. Future satellite-based Earth observation-derived products need to prioritize emerging public health needs as well. This would entail ensuring that the needs of public health related to climate, environmental, and human population changes are met, although these data have multiple applications in the ultimate goal of informing sustainable development and building human resilience.

**Author Contributions:** Conceptualization, S.O.K., C.B. and N.H.O.; methodology, S.O.K., C.B. and N.H.O.; validation, N.H.O.; formal analysis, S.O.K. and N.H.O.; data curation, S.O.K., C.B. and N.H.O.; writing—original draft preparation, S.O.K. and N.H.O.; writing—review and editing, S.O.K.,

C.B., S.B. and N.H.O.; supervision, S.B. and N.H.O. All authors have read and agreed to the published version of the manuscript.

**Funding:** This work was funded by the Public Health Agency of Canada and the Canadian Space Agency under the Climate Change Impacts and Ecosystem Resilience (CCIER) program.

**Institutional Review Board Statement:** Not applicable.

**Informed Consent Statement:** Not applicable.

**Data Availability Statement:** Not applicable.

**Acknowledgments:** The authors wish to thank the following organisations for sharing data on active surveillance of ticks: Public Health Agency of Canada (Zoonotic Diseases and Special Pathogens Division and Centre for Food-Borne, Environmental and Zoonotic Diseases), Manitoba Health, Public Health Ontario, Institut National de Santé Publique du Québec, Université de Montréal and Quebec's Ministère de la Santé et des Services Sociaux, New Brunswick Department of Health, and Nova Scotia Department of Health and Wellness.

**Conflicts of Interest:** The authors declare no conflict of interest.

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
