# Peer review of "Earth Observation-Informed Risk Maps of the Lyme Disease Vector Ixodes scapularis in Central and Eastern Canada"

_remotesensing, doi:10.3390/rs13030524_

Round 1

Reviewer 1 Report

The paper is a successful combination between remote sensing and maximum-entropy approach for modeling species niches and distributions. The advantages of using EO data over ground-level observations are well illustrated in the map and the results are showing a very important practical usage in terms of developing a near-real-time system of risk detection.

Minor suggestion on the maps - you should maintain the same template on the maps (use the same north arrow, map inclination)

Author Response

Response to Reviewer 1 - Specific Comments

Point 1: Minor suggestion on the maps - you should maintain the same template on the maps (use the same north arrow, map inclination)

Rationale 1: We have modified all the maps to make sure that they used the same template in particular the same north arrow, lat/long coordinates and map inclination.

Reviewer 2 Report

Overall, congratulations on the good work.

I have very minor comments/questions: 

Is LULC (line 99) the same as LC (line 96)? Revise the consistent use of terms throughout the document.

Line 136: When stating the ‘meeting’ criteria for some selection, it’s usually understood that the elements had to meet such conditions in order to be part of the subset/selection (in this case selected meteorological stations). The way the 1st criteria is written is rather confusing, as it is written in a way opposite to this general assumption, it explains what was not used (hence, didn’t meet the criteria). I suggest revising items 1 and 2, to share the style of item 3.

Figure 4 and 5. Given that 2800 DD > 0°C is the minimum threshold for the reproduction of tick populations, why wasn’t this value chosen as the break between the classes mapped (instead of breaking at 2825) in Figure 4 and 5?

In Figure 7, how do you explain that the highest slope values (those over 35 degrees Celsius) in Manitoba, Ontario and Quebec, have a geographical proximity to the lowest values (<-10)?

I was concerned by the low AUCROC values obtained. Do you think the decision of treating as "negative" those sites that had historically reported presence and then absence in a subsequent year affected this result? Methodologically speaking, would you suggest any improvements over the method used that could be useful by a follow up analysis or even for the establishment of a continually updating annual monitoring of Ixodes scapularis risk system? 

Author Response

Response to Reviewer 2 - Specific Comments

Point 1: Is LULC (line 119) the same as LC (line 116)? Revise the consistent use of terms throughout the document.

Rationale 1: Yes, we have changed LULC to landcover (LC) to be consistent as suggested. See L119.

Point 2: Line 172-176: When stating the ‘meeting’ criteria for some selection, it’s usually understood that the elements had to meet such conditions in order to be part of the subset/selection (in this case selected meteorological stations). The way the 1st criteria is written is rather confusing, as it is written in a way opposite to this general assumption, it explains what was not used (hence, didn’t meet the criteria). I suggest revising items 1 and 2, to share the style of item 3.

Rationale 2: Yes indeed, we have now modified the text. See L172-176.

Point 3: Figure 4 and 5. Given that 2800 DD > 0°C is the minimum threshold for the reproduction of tick populations, why wasn’t this value chosen as the break between the classes mapped (instead of breaking at 2825) in Figure 4 and 5?

Rationale 3: Yes, the figures are now corrected with a breakpoint at 2800 DD > 0°C. See L326 and L335.

Point 4: In Figure 7, how do you explain that the highest slope values (those over 35 degrees Celsius) in Manitoba, Ontario and Quebec, have a geographical proximity to the lowest values (<-10)?

Rationale 4: In Northern parts of these three provinces, the landscape is mostly dominated by boreal forests and the surface temperatures will fluctuate depending on the extent of the vegetation cover (e.g. the development and abundance of vegetation after a forest fire or forest clear cut can lead to a decrease in the surface temperature of forest habitats. When the vegetation becomes more abundant, the surface temperature as a lower value. This phenomenon could explain the proximity of negative slope values (blue dots: decrease in surface temperature over time) with higher values (red dots: increase in surface temperature over time).

Also, climate is warming faster in the north, even though the north is still climatically colder than in the south (see reference 17).

Point 5: I was concerned by the low AUCROC values obtained. Do you think the decision of treating as "negative" those sites that had historically reported presence and then absence in a subsequent year affected this result? Methodologically speaking, would you suggest any improvements over the method used that could be useful by a follow up analysis or even for the establishment of a continually updating annual monitoring of Ixodes scapularis risk system? 

Rationale 5: We do not think that treating sites that were positive in one year, then negative in the next as “negative”, had a negative impact on the AUC values as it is clear that adventitious ticks can sometimes be collected in field surveillance, and ticks’ populations can clearly establish and then die out in locations with very marginal environmental conditions. Indeed, the way to explore this is via surveillance that systematically revisits previously visited sites, as well as exploring new sites in locations that become climatically suitable as the climate warms. This is expanded upon in the discussion (L430-449).

Reviewer 3 Report

Line 38. It would be helpful for many readers if there were a couple of sentences, explaining that Lyme disease has implications for humans.

Line 123. It’s not clear to me at this point in the paper whether surface degree days (SDD) are counted by adding 1 for each day that there is a positive Celsius temperature, or whether the degrees themselves are summed. It becomes clearer later on when the values per year are in excess of 365, but the definition could be clearer.

Line 144. It seems a bit odd not to make reference to Figure 3.

Line 192. Typo. With should be which.

Line 212. A general point. There are a lot of acronyms in the paper, e.g. AUC, which don’t help with the readability.

Line 229. Typo. “more describe” -> “described”.

A general point. For non Canadian readers, a list of regions and their two-letter abbreviations would be useful. Or at least to explain how to find Nova Scotia. (I assume this is NL, but it’s not obvious.)

Section 3.6. I find this section hard to understand. It would be much clearer if the analysis was in terms of successful Lyme detections, successful non-Lyme detections and false positives and negatives. It’s very easy to make a successful detection algorithm, if you don’t care about false alarms. You can just reduce your threshold so that every pixel counts as a detection, then you have 100% detection, but millions of false positives.

Figure 12. This graph is lacking a key.

Section 4. I don’t think the point being made in this section follows from the analysis.

The maps performed well in validation against surveillance data. Specificity was relatively low, i.e. the maps tended to suggested that more sites should have tick populations in surveillance sites than was observed. This was expected as, because I. scapularis is expanding its range into Canada, not all suitable sites would be expected to have tick populations at the time surveillance took place (and they may now indeed have tick populations; [30]); and because not all woodlands may be suitable for I. scapularis populations (see below).

Basically the argument here looks like “Our algorithm got things wrong, but we expect there are things which haven’t been observed yet, which if they had been observed would have shown our algorithm was correct”. It might be better to look at these regions to see whether there is any evidence that a tick population is emerging here.

Author Response

Response to Reviewer 3 - Specific Comments

Point 1: Line 40-48. It would be helpful for many readers if there were a couple of sentences, explaining that Lyme disease has implications for humans.

Response 1: We have added text to specify Lyme disease’s burden in humans with a Canadian perspective and numbers. See L40-48.

Point 2: Line 146. It’s not clear to me at this point in the paper whether surface degree days (SDD) are counted by adding 1 for each day that there is a positive Celsius temperature, or whether the degrees themselves are summed. It becomes clearer later on when the values per year are in excess of 365, but the definition could be clearer.

Response 2: Yes, the text is now clearer we hope. See L146-149.

Point 3: Line 192. It seems a bit odd not to make reference to Figure 3.

Response 3: Yes, we have changed the text with a reference to Figure 3.

Point 4: Line 259. Typo. With should be which.

Response 4: Text changed.

Point 5: Line 281. A general point. There are a lot of acronyms in the paper, e.g. AUC, which don’t help with the readability.

Response 5: We checked the text and eliminated acronyms that weren't used at least 3 times. Also, we removed any acronym in the conclusion.

Point 6: Line 297. Typo. “more describe” -> “described”.

Response 6: Text changed.

Point 7: A general point. For non Canadian readers, a list of regions and their two-letter abbreviations would be useful. Or at least to explain how to find Nova Scotia. (I assume this is NL, but it’s not obvious.)

Response 7: Yes, this is a good point, we have added an explicit legend under Figure 1, therefore readers will have the meaning of the 2-letter abbreviations for each province/territory showed in the figures. See L359.

Point 8: Section 3.6. I find this section hard to understand. It would be much clearer if the analysis was in terms of successful Lyme detections, successful non-Lyme detections and false positives and negatives. It’s very easy to make a successful detection algorithm, if you don’t care about false alarms. You can just reduce your threshold so that every pixel counts as a detection, then you have 100% detection, but millions of false positives.

Response 8: We are not quite sure we understand this point. First, the sensitivity and specificity values vary according to a selected cut off as in Fig 12, and sensitivity is high and similar up to about 3300 degree days > 0C. Second, as mentioned in the methods (sections 2.5.2 and 2.5.3) the explanatory variable data used for statistical analysis (including the ROC analysis) comprised one value for each surveillance site, and only data from surveillance sites were used in analyses, so we are not using millions of pixel values. However, we have now illustrated what the analysis says and hope that this is what the reviewer was after (L404-413).

Point 9: Figure 12. This graph is lacking a key.

Response 9: A key is now added to figure 12 (Figure 12. Variation in specificity and sensitivity of detection of I. scapularis for different values of the maximum cumulative annual surface degree-days above 0°C within forest covered areas (DDMax). L418.

Point 10: Section 4. I don’t think the point being made in this section follows from the analysis.The maps performed well in validation against surveillance data. Specificity was relatively low, i.e. the maps tended to suggested that more sites should have tick populations in surveillance sites than was observed. This was expected as, because I. scapularis is expanding its range into Canada, not all suitable sites would be expected to have tick populations at the time surveillance took place (and they may now indeed have tick populations; [30]); and because not all woodlands may be suitable for I. scapularis populations (see below).

Basically the argument here looks like “Our algorithm got things wrong, but we expect there are things which haven’t been observed yet, which if they had been observed would have shown our algorithm was correct”. It might be better to look at these regions to see whether there is any evidence that a tick population is emerging here.

Response 10: We have reworded this to make it a bit clearer (L404-413 and L430-449). But it really is expected that the specificity would be low. It would be much more concerning if the algorithm missed positive sites.

Reviewer 4 Report

The authors proposing Earth observation-informed risk maps of the Lyme disease vector Ixodes scapularis in Central and Eastern Canada. They used MODIS LST bands and land cover maps to provide risk maps for I. scapularis growth in central and eastern Canada.

However, the main scientific contribution in Remote Sensing or the novelty of this study is not clear. Fusing LST and land cover maps can be considered a minor contribution. The authors should focus on the novel contribution of their study.

Thus, I suggest rejecting this manuscript.

Moreover, I suggest some recommendations which may help to improve this manuscript. 

Please find below my comments.

  • The selected study period ended from five years which reduce the importance of this study. What is the current or future Remote Sensing tools which can be used in similar studies? I suggest the Landsat 8 thermal bands as updated alternative to MODIS LST bands.
  • The introduction section should include a literature review of the similar studies and the innovation of this study.

Please also consider the following minor comments.

Lines 19-20: Please clarify this sentence.

Line 105: Please follow the journal style for references.

Line 152: Please clarify how this equation was created.

Lines 218-219: Please remove these sentences.

Line 233: Please remove this table and measure the root mean square error and the mean absolute error of this correlation.

Line 237: Please add the axes units to this figure.

Line 246: Pease remove the north direction arrow and you can rotate the maps to the north direction.

Line 263: Please clarify this figure.

Figure 8: Pease remove the north direction arrow and you can rotate the maps to the north direction.

Figure 10: Please clarify this figure.

The discussion section has two references should follow the journal style.

The references should follow the journal references style.

Author Response

Response to Reviewer 4 - Specific Comments

Point 1: The authors proposing Earth observation-informed risk maps of the Lyme disease vector Ixodes scapularis in Central and Eastern Canada. They used MODIS LST bands and land cover maps to provide risk maps for I. scapularis growth in central and eastern Canada.

However, the main scientific contribution in Remote Sensing or the novelty of this study is not clear. Fusing LST and land cover maps can be considered a minor contribution. The authors should focus on the novel contribution of their study.

Response 1: The main contribution of this article is the application of EO data to public health in the context of the effects of climate change on the emergence of an infectious disease. We have added text in the discussion and conclusion, in regards to the contribution (discussion L424-429, L519-533) and the next steps (conclusion L553-559)

The study does not address the fusing (e.g. pansharpening, downscaling) of LST and LC data. It combines the use of these data in order to focus on the surface temperature of forest habitats that could be suitable for tick populations.

The contributions of our study in earth observation and in the application of remote sensing in spatial epidemiology are numerous:

  • Our study proposes a use of earth observation data combining a greater temporal frequency (average of 8 days), better spatial resolution (1000 m and 500 m), a longer period of time (16 years) and a greater geographic coverage (central and eastern Canada) to assess and map the risk of Ixodes scapularis in Canada.
  • The methodology proposed in our study to estimate cumulative surface degree days > 0°C of forest habitats offers better absolute precision of this variable as evidenced by the validation that was done with meteorological data. This variable is rarely validated in studies where it is used to model or assess the risks associated with Lyme disease. We propose in our study an objective method to validate and estimate the precision of annual cumulative surface degree days > 0°C of forest habitats derived from Earth observation images using observational data on field.
  • The method we propose makes it possible to use earth observation images to assess and map the risk of Ixodes scapularis with better precision and better spatial resolution which offers monitoring of the spatial variability of the risk at local scales and better adaptation of control and prevention measures by both health regions and municipalities.
  • In addition to the new "standards" to better estimate annual cumulative surface degree days > 0°C of forest habitats using earth observation images, our study proposes a new indicator derived from the OT, the maximum successive years of positive risk of Ixodes scapularis, to provide information on the intensity and extent of the risk over time. Given the length of the life cycle of I. scapularis and its geographic expansion over time, this new indicator provides information on the persistence of favorable conditions in time and space, and indirectly on the probability that a tick population is established in a given place.

Point 2: The selected study period ended from five years which reduce the importance of this study. What is the current or future Remote Sensing tools which can be used in similar studies? I suggest the Landsat 8 thermal bands as updated alternative to MODIS LST bands.

Response 2: Landsat 8 thermal bands image are acquired once every 16 days. MODIS offer a repetitive data every 5 minutes (weekly or monthly mean). The definition of surface degree days (Surface degree days were estimated as the daily average LST values above 0°C (SDD > 0°C)) and their summation over a period of time (e.g. annual cumulative surface degree days > 0°C) show that Landsat data is not suitable for calculating such a variable.

Point 3: The introduction section should include a literature review of the similar studies and the innovation of this study.

Response 3: We have added text to include the few studies who did similar approach without extensive work since we compiled 16 years (2000 to 2015) of repetitive multi-source data which is unique. See L53-57 and L95-100 in the introduction.

Point 4: Lines 18-19: Please clarify this sentence.

Response 4: We have modified this sentence so now the abstract is easier to understand we hope. L18-19.

Point 5: Line 125: Please follow the journal style for references.

Response 5: Yes, the reference is now changed.

Point 6: Line 202: Please clarify how this equation was created.

Response 6: We hope this is now clear (Line 197-202).

The details and explanations regarding the creation of Equation 2 are provided in reference [27]

Point 7: Lines 288: Please remove these sentences.

Response 7: Yes indeed, these sentences are now removed.

Point 8: Table 1: Please remove this table and measure the root mean square error and the mean absolute error of this correlation.

Response 8: We have added the root mean square error and the mean absolute error of this correlation within the Table 1. We think that table 1 could be kept with these additional measures of error. See L81-189 and Table 1 (L310).

Point 9: Figure 3: Please add the axes units to this figure

Response 9: Figure 3 is now corrected as suggested. See L315.

Point 10: Figure 4: Pease remove the north direction arrow and you can rotate the maps to the north direction.

Response 10: Figure 4 is now corrected as suggested. See L327.

Point 11: Figure 6: Please clarify this figure.

Response 11: We have made a slight change to the legend to this and other figures and hope this is what the reviewer was asking for. See L343.

Point 12: Figure 8: Pease remove the north direction arrow and you can rotate the maps to the north direction.

Response 12: Figure 8 is now corrected as suggested. L367.

Point 13: Figure 10: Please clarify this figure.

Response 13: See response 11 and L373.

Point 14: The discussion section has two references should follow the journal style.

Response 14: Yes, the references are now changed.

Round 2

Reviewer 4 Report

The article has been improved by the authors. Still, I have two minor comments.

First, the literature review needs more comparable studies to demonstrate the importance of this study and its new contribution to the Remote Sensing field.

Second, the selected study period ended from five years ago which considered a weakness point reducing the importance of this study. I suggest adding a section on the possibility of predicting the risk of the Lyme disease for the next five years as example.  

Author Response

Point 1: First, the literature review needs more comparable studies to demonstrate the importance of this study and its new contribution to the Remote Sensing field.

Response 1:

For introduction: We have added two paragraphs and additionnal references to answer these concerns (see L95-128 in the revised manuscript). We hope the contributions are clearer in terms of study importance and in terms of new contribution to the field of "Remote sensing".

Briefly: Previous studies have used EO data to explore spatial variation in environmental suitability, over relatively short timescales, for vector-borne diseases, (including tick-borne diseases), mostly from the perspective of ecological studies. However, here we explore the use of EO data for public health purposes to identify how environmental suitability for the Lyme disease vector I. scapularis has changed over a 15 year timescale when it is thought a warming climate has driven range expansion of the tick and Lyme disease risk into Canada.

Point 2: Second, the selected study period ended from five years ago which considered a weakness point reducing the importance of this study. I suggest adding a section on the possibility of predicting the risk of the Lyme disease for the next five years as example. 

Response 2:

We have added supplemental information in the discussion in regard to this comment, see L572-576:

Here we used data from 2000-2015 to demonstrate the utility of EO data to identify environmental suitability for I. scapularis and thus Lyme disease risk. The method can now be implemented, in concert with ongoing surveillance, to review how Lyme disease risk may have increased in geographic scope year-on-year, and how risk may evolve in coming decades with projected climate change (REF - McPherson M, García-García A, Cuesta-Valero FJ, Belltrami H, Hansen-Ketchum P, MacDougall D, Ogden NH. (2017) Expansion of the Lyme disease vector Ixodes scapularis in Canada inferred from CMIP5 climate projections. Environmental Health Perspectives, 125(5):057008)

-